# Long-Form Speech Generation with Spoken Language Models

Se Jin Park [* 1 2]   Julian Salazar [* 1]   Aren Jansen [1]   Keisuke Kinoshita [1]   Yong Man Ro [2]   RJ Skerry-Ryan [1]

## Abstract

We consider the generative modeling of speech over multiple minutes, a requirement for long-form multimedia generation and audio-native voice assistants. However, textless spoken language models struggle to generate plausible speech past tens of seconds, due to high temporal resolution of speech tokens causing loss of coherence, architectural issues with long-sequence training or extrapolation, and memory costs at inference time. From these considerations we derive **SpeechSSM**, the first speech language model family to learn from and sample long-form spoken audio (e.g., 16 minutes of read or extemporaneous speech) in a single decoding session without text intermediates. SpeechSSMs leverage recent advances in linear-time sequence modeling to greatly surpass current Transformer spoken LMs in coherence and efficiency on multi-minute generations while still matching them at the utterance level. As we found current spoken language evaluations uninformative, especially in this new long-form setting, we also introduce: **LibriSpeech-Long**, a benchmark for long-form speech evaluation; new embedding-based and LLM-judged metrics; and quality measurements over length and time. Speech samples, the LibriSpeech-Long dataset, and any future code or model releases can be found at https://google.github.io/tacotron/publications/speechssm/.

## 1. Introduction

Generative spoken language models (Lakhotia et al., 2021; Dieleman et al., 2021; van den Oord et al., 2017) are autoregressive models of invertible audio representations, enabling the direct learning and generation of intelligible speech and

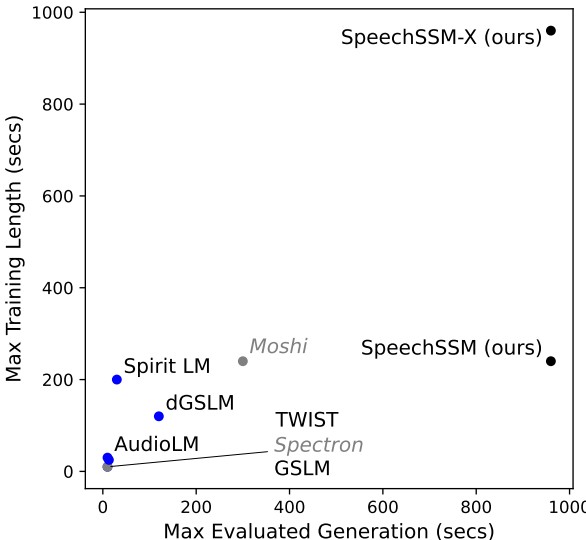

*Figure 1.* Maximum sequence lengths considered by various spoken LMs. *Italicized* models used text intermediates at generation time. Our models can generate indefinitely due to their constant memory footprint, but we cap our evaluations to 16 minutes.

its paralinguistic aspects, such as prosody (Kharitonov et al., 2022) and turn-taking (Nguyen et al., 2023b). These capabilities make speech-native language models (LMs) promising for applications like media understanding and co-creation, audio-native voice assistants, and textless NLP. However, real-world use-cases of spoken LMs require the ability to both understand and generate long-form speech. For example, voice interactions can last many minutes, requiring a model to maintain a growing conversational history in real time, and expressive media like audiobooks and podcasts can require semantic, paralinguistic, and speaker coherence over a chapter or episode.

This presents significant challenges for existing spoken language models, as spoken audio's textual content is entangled with paralinguistic and acoustic properties that detract from learning invariant representations of meaning. Additionally, current audio representations have high temporal rates, requiring 10+ speech tokens to cover the duration of 1-2 text tokens (Hassid et al., 2023). Hence, models must disentangle, aggregate, and generate content coherently over longer time horizons. A single-stage Transformer (Vaswani et al., 2017) LM is impractical, as its initial cost grows quadrati-

---
[*]Primary contributors. Se Jin's work was done as a student researcher at Google DeepMind. [1]Google DeepMind. [2]Integrated Vision and Language Lab, KAIST. Correspondence to: Se Jin Park <jinny960812@kaist.ac.kr>, Julian Salazar <julsal@google.com>.

*Proceedings of the $42^{nd}$ International Conference on Machine Learning*, Vancouver, Canada. PMLR 267, 2025. Copyright 2025 by the author(s).

**Prompt (ASR):** "Oh, it's easy enough to make a fortune," Henry said. "It seems to be easier than it is, I begin to think," replied Philip.

**SpeechSSM Continuation (ASR):**
"Well, if you've got your fortune in your head, it's not that yet," Philip said. "But you needn't be afraid of a Desert Mountain, or a Rosalind Coral Mountain, or with your name in the book, you know. When you mention that fortune, your head will get disturbed, and you'll find you have all sorts of pictures which give you a dinner time."

There was no telling what they had disturbed the day before. Philip, who had a tendency to see the failures of travel with a good deal of ambition, and a perfectly happy way, too, moved over to the gate.

The great buildings resembled the pile of beer, with the little river fronts hanging down in length before them. The weather temperature also, was very transversally clear. It would seem as if Philip had laid out a pair of scissors, and a little box, and tried to pull the panel and make money. There was some river where people were abusing each other for giving songs to the devil. And such a body of music had been given by Horace Barrows, who had come in from the world of Grossi and Tortel, to sing wonderful songs. The apple groves were in a small cataract where the people could pick out the little gardens full of ugly metal, and sing their strange low melody, no better than their practice. "I don't think there's the nearest aerial to the traveling song to do," said Philip.

"I didn't hear the music long ago, for I learned it in such tones," replied Horace Barrows.

*(more speech...)*
It was four o'clock, when Pierre called Prince Albert and explained what had happened. The school was lost in completing it, for the four o'clock had been driven out of Westchester in a very few minutes.

At last Philip was exhausted. He didn't hear a word, but entered the coach and was driven by Prince Albert into the street. "I have some news to discuss. Captain Kidd will come for you presently," was the cordial response of the Princess Maria Theresa, who was still in her green satin. They all talked gallantly, and Philip believed that it was a genuine disaster. "They would have to take him for supper, and Mr. Phineas, I suppose you wanted to join the sport. I told you he was a handsome fellow," said Philip smiling frankly at the boy.

*(more speech...)*
So Philip and Princess Maria went to look for him in the Dragon Springs garden, and there most of the guests were gone. Philip was terribly excited.

"They must get some of the guests in the kitchen," exclaimed the guests, selecting a bright fire in the bright, sunny spot. Philip had never felt so happy in his life before. "Did you hear about the Dragon Springs?" he asked, after a pause. "Yes, they say that when they were strong enough, the famous Albanians, British and Germans, were worthy

**SpiritLM Continuation (ASR):**
Well, you'll see. said his brother, getting in his hat and coat.
Now here's the second cup.
Here's the second cup, unless you'd bear, take it. Huh.
Okay, pay five helper.
Go on, set down now, and now, this time, and don't say is here. Well. Depend on yourself. Alright. Here for quick off, two pence.
And with the thrust of his waist, to hasten his way, quick, he went off. But then he'll come back, top into again, well much to get back, Philip.

He filled the cup he was thought, he felt little cup, with much made thick thought, from bra. Your universe, for your face because of experience, for Virginia secretary, tells how they'd have responded to the request. Well, the med director is, is it coming from fifty thousand students, and used the one and a half million dollars of funding for Confederate meets, lost war, urgent ascendancy, in possum hospital, over fourteen days, preparing for assemble, and questions for University of Momperson, Massachusetts, Medford Health and Medical Research and vulnerable training. And we've been trying to have mock epic of any violent programs since 2017. Their for university and transfer program. For the simple part, freshly from droppin'.

Great. Possible photograph from
[Laughter]
This is PTSD. This likes me. This is great. This is great. This is great. This is what. This is great. This is great. This is great. This is great. This is great. This is great. This is great. This is great. This is great. This is great. This is great. This is great. This is great. This is great. This is great. This is great. This is great. This is great. This is great. This is meant. This is great. This is great. This is great. This is great. This is it. This is great. This is great. This is great. This is it. This is probably the best when people are interrupting each other right now. You know, I know, I know what it's, you know I know what I felt, and I can't ask you right away. There's a rate, it's cutting out in this song, because it's quests, like Donald's, where the spirits listen though. Although, other stuff, I hate doesn't just focus on artists. Perhaps putters. It's bait. It's bait, it's a bait, it says. Actually, he said that before,

*(more speech...)*
It's going to be. It's going to be. It's going to be. It's going to be. You can see it tonight. It's going to be. It's going to be. So freaking. So fun. So fricking fun. So fun. So fricking fun. So fricking. So fricking. So fricking. So fricking. So fricking. So fricking. So fricking. So fricking. So fricking. So fricking. So fricking. So fricking. So fricking. So fricking. So fricking. So fricking. So fricking. So fricking. So fricking. So fricking. So fricking. So fricking. So fricking. So fricking. So fricking. So fricking. So fricking. So fricking. So fricking.

*Figure 2.* Automated transcripts of 4min speech continuations generated by SpeechSSM-2B (ours) and a Spirit LM Expressive (7B) model (Nguyen et al., 2025) under slide-and-prompt generation (Section 7), extending a 10-second audio-only prompt from our proposed LibriSpeech-Long (test-clean). Aspects like recurring proper nouns show SpeechSSM's relative semantic consistency over time.

cally with prompt length, and its per-step cost grows linearly when decoding. Furthermore, it may also be ineffective, as suggested by Transformer's degraded performance on long-range tasks (Tay et al., 2021). Though a few works have improved speech coherence via joint modeling with text (Section 2), the challenge of directly modeling long-form speech, particularly generation, remains unstudied by existing spoken LMs (Figure 1) and the field overall. Our work proposes and makes initial progress on generative long-form speech:

**Modeling.** We discuss the design choices required to enable the practical training, generation, and extrapolation to tens of minutes of audio, from tokenization to speaker conditioning to complexity with respect to sequence length. The result is **SpeechSSM**, a new (textless) spoken language model family (2B, 9B) designed for long-form generation, being the first to model and generate unbounded long-form speech in bounded memory and the first state-space spoken LM. As baselines, we also train spoken Transformer LMs to perform multi-minute generations. Finally, we demonstrate **SpeechSSM-X**, an extemporaneous variant for naturalistic spontaneous speech.

**Evaluation.** We observe that existing metrics in speech gen-

eration evaluation are noisy and poorly discriminative, and propose the use of reference-based semantic metrics, side-by-side LLM-as-judge, and time-stratified evaluations for speech generation. To scale these to long-form evaluation, we introduce the **LibriSpeech-Long** benchmark, which reprocesses LibriSpeech's (Panayotov et al., 2015) dev and test sets' original chapter-level audio into utterance-aligned spans of up to 4 minutes. This enables much longer prompts and ground truths for reference-based evaluations in tasks like long-form speech continuation, speech recognition, and text to speech.

We find that SpeechSSM matches existing spoken LMs on short generations, while outperforming their sliding window extensions on long generations (e.g., Figure 2), and that our proposed metrics and benchmark distinctly quantify the quality gaps between past work, our work, and human-level speech generation—enabling future development. We release examples of read- and extemporaneous-style generations of up to 16 minutes in length,[1] and we release the LibriSpeech-Long evaluation dataset[2] under a CC-BY 4.0 license.

---

[1] https://google.github.io/tacotron/publications/speechssm/
[2] https://github.com/google-deepmind/librispeech-long/

## 2. Related Works

**Generating with Spoken LMs.** The family of GSLM models (Lakhotia et al., 2021; Kharitonov et al., 2022) are Transformer decoder LMs trained on discrete units obtained from $k$-means clustering of HuBERT (Hsu et al., 2021) features and synthesized via unit-to-spectrogram or unit-to-waveform models. This approach gave promising temporal coherence but poor audio quality, and so AudioLM (Borsos et al., 2023a) proposed separate LMs, one for semantic tokens as before, and two for modeling coarse-to-fine acoustic tokens that are residual codes of a neural audio codec (Zeghidour et al., 2022); this was simplified and made non-autoregressive by Borsos et al. (2023b). TWIST (Hassid et al., 2023) found that text LM initialization improved content-level coherence, atop which VoxtLM (Maiti et al., 2024) and Spirit LM (Nguyen et al., 2025) found that joint or interleaved training with text gave further improvements.

Beyond the scope of this work, there are audio-text models like SpeechGPT (Zhang et al., 2023a) trained for sequence-to-sequence and not generative continuation; there are also dual-channel and text-intermediate models like dGSLM (Nguyen et al., 2023b) whose semantic evaluations are <20s, Spectron (Nachmani et al., 2024) which passes through text, and Moshi (Défossez et al., 2024) which had few-minute dialogues via time-aligned text.

**State-Space Models for Long-Form Audio.** State-space models (SSMs; Gu et al., 2021) have become popular among efficient (sub-quadratic) replacements for Transformer-based architectures, giving the first model (S4; Gu et al., 2022) to perform all tasks in the Long-Range Arena (Tay et al., 2021), outperforming the vanilla Transformer. They utilize constant computation and memory requirements to generate tokens during inference and can be efficiently trained. Recent focus has shifted to hybrid models (Glorioso et al., 2024; Lenz et al., 2025; De et al., 2024) which integrate state-space layers and variants like linear recurrent units (LRU; Orvieto et al., 2023) with finite-context self-attention layers. Recent works have considered SSMs in audio, primarily to support long speech *inputs* for text-out tasks like automatic speech recognition (ASR) and summarization. None are spoken LMs for speech continuation, with only one considering (acoustic-level) tokens (Gao & Chen, 2024); most works involve spectrogram encoders or outputs (Shams et al., 2024; Erol et al., 2024; Lin & Hu, 2024; Miyazaki et al., 2024). Closest in spirit is SaShiMi (Goel et al., 2022), a multi-scale S4 operating directly on waveform samples; though they generated only 1s of speech, this corresponds to a sequence of 16k discretized scalars.

**Evaluating Spoken LM Generations.** Lakhotia et al. (2021) was first to evaluate the generations of spoken LMs, proposing ASR as a path to automated text metrics like text perplexity (PPL) and proportion of repeated $k$-grams (auto-BLEU), along with human evaluations of intelligibility and meaningfulness with mean opinion scores (MOS and MMOS respectively). For their spoken LMs, zero-shot (non-generative) metrics based on logprobs of contrastive pairs (sWUGGY and sBLIMP; Nguyen et al., 2020) were predictive of generation performance, though scores varied with token vocabulary size. However, these initial metrics seem to lack robustness or are saturating with respect to newer spoken LMs. Hassid et al. (2023) found transcript PPL and auto-BLEU to be noisy, favoring MMOS and expanding zero-shot metrics (sStoryCloze and tStoryCloze). In turn, Défossez et al. (2024) found that sWUGGY and sBLIMP scores degraded despite experiential improvement from noise augmentation and instruction-tuning, instead favoring spoken question-answering (Nachmani et al., 2024) evaluated via ASR.

Closest to our work was the use of LLMs to assign absolute, reference-free scores to assess the instruction-following of turn-based text-and-speech LMs in Zhang et al. (2023a; 2024). As for observing saturation, Borsos et al. (2023a) found that humans could not distinguish between a synthetic 7s continuation versus the real 7s continuation of a 3s prompt on a holistic side-by-side evaluation, suggesting the need for more targeted and longer-form evaluations.

## 3. Unbounded Speech Generation

We begin by proposing a set of requirements for a general, unbounded, speech generation system:

- **Constant memory during decoding**, to enable indefinite AR sampling without running out of memory.

- **Infinite context**, so that arbitrarily distant dependencies can be expressed, at least in theory. With the above, this means relevant context must fit in a fixed-size state.

- **Generative length extrapolation**, so that speech quality remains consistent over time, in particular beyond audio durations seen during training.

The first leads us to linear-complexity sequence modeling with a fixed-size state. The second leads us to models with aggregation mechanisms such as recurrences or compressive memories. We show that with some care, one can also achieve the third requirement of generative extrapolation.

Finally, there is also a soft requirement for *efficient training*, e.g., train-time dependence on sequence length that is sub-quadratic, to enable longer sequences and reduce reliance on extrapolation. This favors a parallelizable weight learning scheme, which naturally leads to state-space models (broadly defined, i.e., including linear recurrence models and certain hybrid variants; Patro & Agneeswaran, 2024; Dao & Gu, 2024) and thus SpeechSSM, a family of hybrid

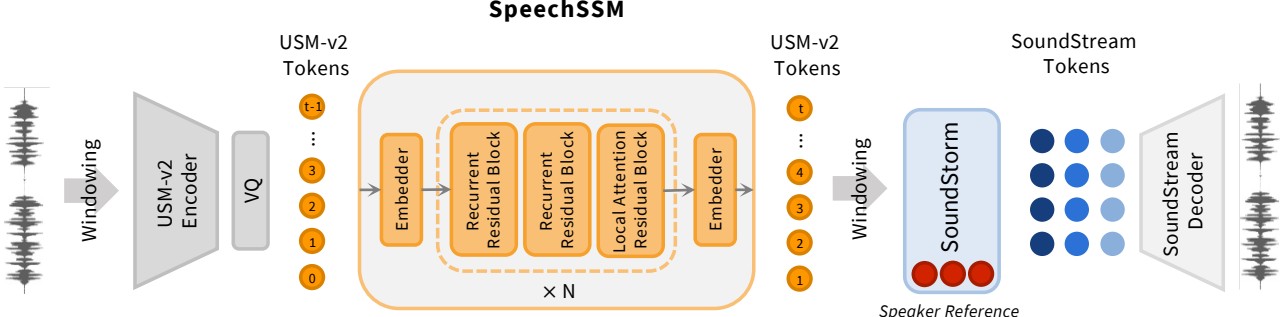

*Figure 3.* Overview of SpeechSSM. *Left:* A causally-masked hybrid state-space model (Griffin) is trained with an LM objective on semantic tokens (USM-v2) encoded via overlapping fixed-size windows. *Right:* A non-autoregressive synthesizer (SoundStorm) converts overlapping windows of semantic tokens to the acoustic tokens of a neural codec (SoundStream) in a speaker-conditioned manner.

state-space spoken language models for efficient long-form speech generation that fulfills all these desiderata:

**Architecture.** For our decoder-only hybrid SSM we choose Griffin (De et al., 2024), which interleaves a gated variant of LRUs (Orvieto et al., 2023) and local (sliding-window) multi-query attention (MQA) blocks in a fixed pattern (two recurrent, one local-MQA; see Figure 3, left). Local attention efficiently captures recent context, while the states of the gated recurrences transmit information across arbitrary distances. Griffin's performance matched comparable Transformers while greatly improving inference speed and enabling context-side extrapolation at least 4x longer than seen in training. As RoPE (Su et al., 2024) in the local-MQA blocks still encodes absolute position, we follow recent work on position embeddings (PEs) under causal self-attention (NoPE; Kazemnejad et al., 2023) and remove all explicit PEs from SpeechSSM to promote extrapolation.

**Initialization.** Inspired by Hassid et al. (2023)'s success with text-initialized spoken language models (TWIST), we initialize our models with RecurrentGemma-{2B,9B} IT (Botev et al., 2024), which are open-weight LMs with the Griffin architecture, trained on 2 trillion text tokens. We discard the pretrained text token embeddings and initialize new ones for our audio token vocabulary.

**Semantic Tokenizer.** We use the pretrained USM-v2 speech tokenizer (Vashishth et al., 2024; Rubenstein et al., 2023). Its encoder (Zhang et al., 2023b) is trained with masked language modeling on untranscribed audio and an auxiliary ASR loss on transcribed audio. Inner representations are vector-quantized into 32k units that serve as fixed-rate (25Hz) pseudo-text for our speech LM. Vashishth et al. (2024) found that USM-v2 was by far the most speaker-invariant token among common speech tokenizers.

**Speaker-Prompted Audio Synthesis.** Following Borsos et al. (2023a), we have a second stage that generates low-level acoustic tokens conditioned on semantic tokens. We

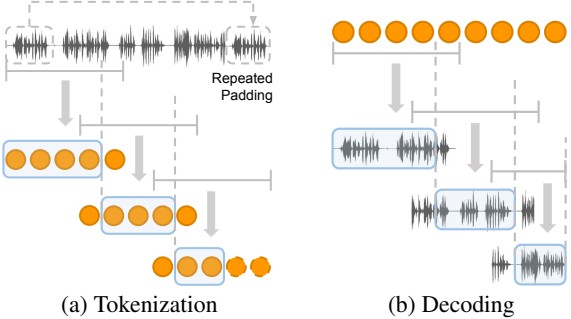

| (a) Tokenization | (b) Decoding |
| --- | --- |

*Figure 4.* Depiction of how input and output windowing work, shown here with 5-token window widths and 2-token overlaps.

use a SoundStorm model (Borsos et al., 2023b) to non-autoregressively generate SoundStream tokens (Zeghidour et al., 2022), a standard neural audio codec that efficiently reconstructs to high-quality audio. Notably, one can train SoundStorm to support 3s voice prompts (a frozen prefix of semantic and acoustic tokens), such that output acoustic tokens reflect its speaker characteristics.

By choosing a token and model decomposition that isolates speaker characteristics to the acoustic stage, SpeechSSM focuses capacity on modeling semantic coherence along the temporal axis.

**Windowed Tokenization and Decoding.** To process long-form speech while bounding the memory of the (non-SSM) semantic tokenizer and acoustic decoder, we divide audio into fixed segments, with overlaps between neighbors. Each window is tokenized independently, then merged into a single stream at each overlap by taking the first half's tokens from one window and the second half's tokens from its successor (Figure 4a). For synthesis, fixed token windows, each conditioned on a short speaker prompt (3s), are synthesized into waveform independently then merged with the same boundary overlap adjustment (Figure 4b). We find these strategies minimize boundary artifacts while enabling continuous tokenization and decoding over time.

**Avoiding Implicit EOSes.** Despite having no end-of-sequence (EOS) tokens, our early models did not generatively extrapolate (e.g., a 4min model reaching 4.5min before degrading to noise/silence). In non-causal semantic tokenizers like USM-v2, we found the remaining length-in-window may be implicitly encoded in tokens, making tokens in final windows look "different." As evidence, padding the last window to 30s using silence, tokenizing, then dropping those tokens led to silence, as "future" silence was now in the kept tokens. What worked was (1) to pad the last window to 30s using speech from the beginning of the example (depicted in Figure 4a) so that final tokens were tokenized as if there was further speech, and (2) in the case of Libri-Light, to still drop the last 10s of examples—as the padded beginnings were disproportionately "Chapter *<number>*"!

## 4. Improved Evaluations for Spoken LMs

**Updated NLG Evaluations.** The shortcomings found in recent work (Section 2) align with recent developments in natural language generation (NLG) evaluation, which have moved beyond intrinsic and/or surface word metrics like PPL, auto-BLEU, self-BLEU (Zhu et al., 2018), especially for open-ended generation. One major shift has been the adoption of **embedding-based metrics**, where distances are computed between embeddings of generated versus reference text (Sai et al., 2022). A more recent trend uses instruction-tuned LLMs to perform automated Likert-scaled evaluations (Li et al., 2024b), as applied by Zhang et al. (2023a; 2024) to text-instructed speech generation and could be extended to speech as well.

However, to tackle the saturation of acoustic evaluations and to leverage text references, we in particular propose **automated side-by-sides** (LLM-as-a-Judge; Zheng et al., 2023) of total transcripts to scalably compare systems against the ground truth and each other. This has particular advantages for spoken LMs: (1) It mitigates the noise from ASR issues highlighted by Hassid et al. (2023), as the compared generations will be both afflicted (the *ground truth should also be re-transcribed by ASR*, for fairness). (2) It works around the subtle issue of fixed-duration slices occurring mid-word, degrading PPLs; instead, one can always transcribe prompt and continuation together, leaving the LLM to focus on the contrast. We implement both proposed evaluation types in Sections 6.2 and 7.1.

**LibriSpeech-Long.** To extend reference-based metrics to long-form speech, one needs long-form reference speech and transcripts. Although 3s prompts from LibriSpeech (Panayotov et al., 2015) have been the standard benchmark for spoken LMs since Lakhotia et al. (2021), their provided references have an average length of 10s, making them unsuitable beyond 7-10s continuations. Observing that the LibriSpeech dev and test sets are derived from full chapters

*Table 1.* Statistics of our proposed LibriSpeech-Long benchmark, which was generated with a maximum target duration of 4 minutes.

| Subset | # Hours | # Examples | Avg. Dur. (s) | # Chapters | # Spkrs |
|---|---|---|---|---|---|
| dev-clean | 16.0 | 295 | 194.8 | 97 | 40 |
| dev-other | 9.5 | 188 | 182.4 | 91 | 33 |
| test-clean | 14.8 | 270 | 197.6 | 87 | 40 |
| *>3.5min* | *12.6* | *193* | *234.2* | *82* | *40* |
| test-other | 10.7 | 207 | 185.9 | 90 | 33 |
| *>3.5min* | *8.2* | *126* | *234.4* | *77* | *32* |

of public-domain audiobooks which have been excluded from the standard LibriLight training set (Kahn et al., 2020) used by many spoken LMs, we reprocessed their source uncut audio files, similar to the convenience script[3] in Libri-Light which agglomerates utterances up to a target length of 4 minutes (240s) along utterance boundaries. The results enable longer prompts and references for our side-by-side and similarity metrics. Statistics are shown in Table 1; 64%–76% of each split's examples are >3min long.

**Generation Quality over Time.** While ASR can capture degenerate cases like repeated words (Figure 2), we found that it can fail on cases exacerbated by long-form generation, such as extended silences and voiced non-speech, suggesting the continued importance of audio-native evaluations. Furthermore, we find that generation failures generally increase as decoding progresses over time, which we must quantify to determine if our model has the desired property of generative length extrapolation (Section 3). Towards this, we propose computing **semantic and acoustic metrics that are stratified over the decoding process**. This progression can be expressed in terms of semantic content (number of words into the ASR transcript), or acoustic content (time offset into the generated speech). We describe our implementations of both in Section 7.2.

## 5. Experimental Setup

**Training and Generation.** Following Borsos et al. (2023a), Nachmani et al. (2024), and others, we train on the unlab-60k split from LibriLight (Kahn et al., 2020). Unlike prior work, we study the effect of sequence length on long-form generation, segmenting the audiobooks into training sequences of up to a target duration. The default for **SpeechSSM** is 4min (240s) during training, though we compare with target durations of 30s and 16min (960s) as well ("*with 30s/16min segments*"). We train **-2B** and **-9B** variants, corresponding to RecurrentGemma. Each model is trained with 16 TPUs (v5p) and data parallelism for 100k steps with 768k tokens per batch, and a checkpoint is chosen via transcript perplexity on LibriSpeech-Long dev-clean; more details in Appendix B.1. We sample semantic (USM-v2) tokens with temperature 1. Then the SoundStorm model

---

[3] libri-light/data_preparation/cut_by_vad.py on GitHub.

*Table 2.* Results on short-form generation on LibriSpeech test-clean. Generations are 7s continuations of 3s prompts. **Bolded** are ours. Text-PT, FT denote pretraining (via LM init.) and finetuning with text. Win%$_{GT}$ denotes the win rate of the model over the ground truth. $|V_{audio}|$ denotes speech token vocabulary size. For naturalness mean opinion score (N-MOS) we report 99% confidence intervals.

| Method | Text-Based (ASR) | | | | | Speech-Based | | | | |
| --- | --- | --- | --- | --- | --- | --- | --- | --- | --- | --- |
| | *Text-PT* | *Text-FT* | *PPL↓* | *SBERT↑* | *Win%$_{GT}$ ↑* | $|V_{audio}|$ | *SpkrSim↑* | *N-MOS↑* | *sWUGGY↑* | *sBLiMP↑* |
| GSLM (0.2B) | ✗ | ✗ | 6.28 | 0.17 | 1.4 | 100 | 0.36 | 2.23 ± 0.11 | 64.8 | 54.2 |
| AudioLM (0.9B) | ✗ | ✗ | – | – | – | 1k | – | – | 71.5 | **64.7** |
| TWIST-1.3B | ✓ | ✗ | 7.25 | 0.18 | 3.6 | 500 | 0.41 | 3.09 ± 0.12 | 72.7 | 57.0 |
| TWIST-7B | ✓ | ✗ | 6.54 | 0.20 | **15.5** | 500 | 0.41 | 3.24 ± 0.13 | **73.9** | 59.0 |
| VoxLM (1.3B) | ✓ | ✓ | – | – | – | 200 | – | – | 65.6 | 57.1 |
| Spirit LM Expressive (7B) | ✓ | ✓ | 6.17 | 0.19 | 7.7 | 665 | 0.45 | 3.00 ± 0.08 | 65.0 | 54.2 |
| **SpeechSSM-2B** | ✓ | ✗ | 5.76 | **0.23** | 7.9 | 32k | **0.79** | 3.87 ± 0.07 | 55.8 | 60.9 |
| *without LM init.* | ✗ | ✗ | 6.15 | **0.23** | 7.7 | 32k | **0.79** | 3.80 ± 0.07 | – | – |
| *with Transformer instead* | ✓ | ✗ | 6.16 | 0.22 | 8.4 | 32k | **0.79** | 3.74 ± 0.08 | – | – |
| *with 30s segs. instead* | ✓ | ✗ | 5.73 | 0.22 | 10.5 | 32k | **0.79** | 3.84 ± 0.08 | 57.3 | 61.1 |
| *with 16min segs. instead* | ✓ | ✗ | 5.84 | 0.20 | 4.0 | 32k | **0.79** | 3.86 ± 0.07 | 54.3 | 60.4 |
| **SpeechSSM-9B** | ✓ | ✗ | **5.60** | **0.23** | 13.5 | 32k | **0.79** | **3.94 ± 0.07** | – | – |
| *Ground Truth* | *–* | *–* | *5.63* | *1.00* | *50.0* | *–* | *0.84* | *4.02 ± 0.07* | *–* | *–* |

(speaker-prompted with the first 3 seconds of the prompt) and windowing (30s with 4s overlaps) give acoustic (Sound-Stream) tokens, which a SoundStream codec turns into waveform; model details for these are in Appendix B.2.

**Baselines.** We compare SpeechSSM with decoder-only speech LMs. We use **GSLM** (Lakhotia et al., 2021)'s best model (HuBERT-L6 tokens with vocab 200), trained on the clean 6k hours of LibriLight. For **TWIST** (Hassid et al., 2023), we use both the OPT-1.3B and LLaMA-7B versions, trained on 150k speech hours. For the 7B **Spirit LM** (Nguyen et al., 2025), we use the **Expressive** version model which adds expressiveness via pitch and style tokens in addition to HuBERT tokens. We also cite numbers from **AudioLM** (Borsos et al., 2023a) and **VoxLM** (Maiti et al., 2024), which are both 2B models trained on unlab-60k. VoxLM and Spirit LM see text data during training. Due to variations in token, initialization, and training data choices, we also define **SpeechTransformer** ("*with Transformer*"), a spoken LM initialized with Gemma-2B (Gemma Team et al., 2024) but otherwise matched with SpeechSSM-2B.[4]

## 6. Short-Form Continuation Experiments

Before considering long-form, we compare SpeechSSM to its Transformer-based counterparts as in past work. This takes 3s prefixes from LibriSpeech's test-clean set and generates 7s continuations, which one then transcribes with an ASR model (our details in Appendix B.3). We used test examples with ground-truth continuations ≥5s.

### 6.1. Existing Metrics and Results

**Transcript Perplexity (PPL):** As in prior work, we compute the log-perplexity of the transcript of the generated

continuation under Gemma-2B, as an initial proxy for content fluency.

**Speaker Similarity (SpkrSim):** To analyze voice preservation, we speaker-embed both the prompt and its generated continuation and compute their cosine similarity. We use a speaker classifier used in AudioLM (Borsos et al., 2023a) as the speaker embedder.

**Naturalness Mean Opinion Score (N-MOS):** We evaluate how natural the speech sounds, ignoring the grammar and content of the utterance; this focuses attention on issues not visible on transcripts, ranging from synthesis issues and unintelligible speech through to coherent but unnatural prosody; rater details in Appendix C.

**sWUGGY and sBLiMP:** These probe if the spoken language model can implicitly perform lexical and syntactic contrasts (Nguyen et al., 2020).[5] One reports the % of time the model's log-likelihood ranks the semantic tokens of a correct utterance versus its incorrect counterpart.

Our models' continuations (Table 2) are most speaker-similar to the prompt, which we attribute to our speaker-promptable acoustic stage; in contrast, GSLM and TWIST can only propagate speaker identity via their semantic tokens (plus coarse style and pitch tokens in Spirit LM Expressive). Our N-MOS scores suggest high semantic-to-audio synthesis quality, which we attribute to USM-v2's large vocabulary (32k) and the staged approach via existing codec (SoundStream). SpeechSSM's naturalness is on par or better than a comparable Transformer and close to real speech; see supplementary. Meanwhile, our sWUGGY scores are much worse, while our sBLiMP scores are neutral to above-

---

[4]Note that Gemma saw 50% more text than RecurrentGemma.

[5]sWUGGY: audio pairs of a real word and a fake but similar-sounding word. sBLIMP: audio pairs of a syntactically correct sentence and an incorrect one.

average. These do not positively correlate with text-based scores or subjective quality; instead, they match Lakhotia et al. (2021)'s observations and Borsos et al. (2023a)'s Figure 2, where sWUGGY scores hit relatively sharp maxima at vocabularies of a few hundred tokens. We argue increased noise is expected with larger vocabularies, as then tokens of an audio utterance represents less of the probability mass of all possible renditions of its text. Hence, separately from Défossez et al. (2024), we move towards transcript-based evaluations, as these *marginalize* over spoken renditions to get less noisy evaluations. Finally, even the 2B SpeechSSMs had the lowest transcript perplexities, which is surprising given the other models are larger (7B) and/or have jointly trained with text, and one would not expect SSMs to confer a semantic edge in the $\leq$10s speech horizon. We note that Lakhotia et al. (2021) already caveated—and Hassid et al. (2023) actively discouraged—the use of ASR PPL given its sensitivity to e.g. audio sampling temperature; such metrics may simply indicate model repetitiveness at default settings.

### 6.2. Proposed Metrics and Results

The above points—noise from contrastive audio probes, saturation of N-MOS and SpkrSim, suspicious results from transcript perplexity—**all motivate our proposed shift to newer, reference-based NLG metrics** (Section 4):

**Semantic Similarity (SBERT).** We measure the distance between the semantic embedding of the transcriptions of the generated speech and the reference, using Sentence-BERT MiniLM-L6-v2 (Reimers & Gurevych, 2019) as the semantic embedder. This expresses contextual alignment between the generated text to the ground truth, focusing on semantic meaning over surface-form patterns.

**Side-by-Side Win Rates (Win%$_{\text{vs. model}}$).** We ask the model to analyze then rate (Chiang & Lee, 2023), forking the format of Arena-Hard-Auto's LLM-Judge System Instruction (Li et al., 2024a). Given the book domain and the relatively unconstrained nature of speech continuation, we base our criteria on questionnaires from story generation evaluation (Xie et al., 2023) around *fluency*, *coherence*, *logicality*, and *interestingness*; see Appendix D for the template. We evaluate each prompt twice with order of presentation flipped. Gemini 2.0 Flash (Gemini Team et al., 2025) re-transcribes both model and true audio (without windowing and jointly with prefix) and performs judgments.

Benefits are evident in side-by-sides versus a transcript of the ground truth, with results (Table 2) now matching qualitative experience and expectations (larger models performing better, with Spirit LM Expressive underperforming TWIST perhaps due to capacity spent on pitch/style). However, even SpeechSSM-9B and TWIST-7B win <20% versus transcribed ground truth, suggesting that (automated) **side-by-side comparison on transcripts is more discrim-** **inative than a holistic human side-by-side audio task** in selecting the synthetic sample, where humans performed at random in Borsos et al. (2023a), as it focuses on the content of the speech rather than superficial naturalness. Though our model is not the most fluent in this regime, SBERT foreshadows benefits in faithfulness (our models' continuations are semantically closest to the true ones).

## 7. Long-Form Generation Experiments

We conduct the first evaluation of long-form speech generation, taking extended prompts of 10s from our proposed LibriSpeech-Long (test-clean) and having each model continue them through to 4 and 16 minutes. As other off-the-shelf models trained on sequences well below 4min and were not designed to generate beyond their training length (e.g., use of position encodings), we found them unable to generate beyond a minute without being stuck in noise or silence. To give functional baselines, we propose applying *slide-and-prompt* generation (Borsos et al., 2023b) to the semantic LM itself; that is, we generate to each model's maximum completion length (Figure 1) first, and then repeat generation using the last 3s of the previous window as context. Example generations are in Appendix E.

### 7.1. Semantic and Acoustic Results

We again measure existing and our proposed metrics, with two key changes: first, *we replace Sentence-BERT with Gecko* (Lee et al., 2024) as embedder for semantic similarity as Sentence-BERT's 512-token context cannot handle the transcripts of 4min+ generations; more long-form evaluation details in Appendix B.3. Second, extrapolation failures cause e.g., GSLM and TWIST to generate far less than our models; for fairness *in side-by-sides, we truncate to the shorter transcript's length.*[6]

Our results are in Table 3. The SpkrSims of SpeechTransformer, SpeechSSM, and ground truth increased from 0.79 in short-form to 0.85, likely from increased confidence from longer prompt and continuations; all other decreased. This shows the advantage of our more speaker-invariant USM-v2 tokens and a speaker-prompted audio stage; identity is modulated by the semantic-to-acoustic model, instead of consuming capacity, and being imperfectly carried by, the semantic LM and windowing. SpeechSSM has the best ASR PPL and wins a majority of time vs. all models; our 2B Transformer variant is close at 30sec, but not at 16min (Table 4) —recall that 16min is *24,000* USM-v2 tokens!

However, models achieve *zero wins over the ground truth*. As generation length increases, faults in fluency, coher-

---

[6]Note this means the *relative order* of models in a single Win%$_{\text{vs. model}}$ column should not be read into too much, as challenger models induce truncation to very different lengths.

*Table 3.* Evaluations on LibriSpeech-Long test-clean. ⊞ denotes model extension via windowed generation. Generations are 4m completions of the 10s prompt. Win%$_\text{SSM-2B}$ and Win%$_\text{GT}$ are model wins versus SpeechSSM-2B and ground truths (>3.5min) respectively. For naturalness MOS over time (N-MOS-$T$), the same 5s time span is sampled over all models from each minute of completion.

| Method | Text-Based (ASR) | | | | Speech-Based | | | | |
|---|---|---|---|---|---|---|---|---|---|
| | PPL↓ | Gecko↑ | Win%$_\text{SSM-2B}$↑ | Win%$_\text{GT}$↑ | SpkrSim↑ | N-MOS-$T$↑ (99% CI) | | | |
| | | | | | | <1min | 1-2min | 2-3min | ≥3min |
| GSLM (0.2B)⊞ | 4.74 | 0.67 | 17.6 | 0.0 | 0.33 | 2.00 ± 0.22 | 2.01 ± 0.23 | 1.93 ± 0.21 | 2.09 ± 0.22 |
| TWIST-1.3B⊞ | 5.70 | 0.60 | 16.2 | 0.0 | 0.38 | 1.96 ± 0.22 | 1.59 ± 0.13 | 1.40 ± 0.09 | 1.36 ± 0.09 |
| TWIST-7B⊞ | 4.93 | 0.65 | 24.0 | 0.0 | 0.37 | 2.43 ± 0.27 | 2.15 ± 0.22 | 1.83 ± 0.20 | 1.84 ± 0.18 |
| Spirit LM Expressive (7B) | 5.71 | 0.63 | 24.2 | 0.0 | 0.41 | 2.77 ± 0.12 | 2.52 ± 0.12 | 2.56 ± 0.11 | 2.60 ± 0.10 |
| **SpeechSSM-2B** | 3.75 | 0.70 | *50.0* | 0.0 | **0.85** | **4.12 ± 0.08** | **4.13 ± 0.06** | **4.13 ± 0.07** | **4.16 ± 0.08** |
| *without LM init.* | 4.57 | **0.71** | 42.4 | 0.0 | **0.85** | 3.76 ± 0.09 | 3.76 ± 0.10 | 3.71 ± 0.10 | 3.80 ± 0.08 |
| *with Transformer instead* | 4.71 | **0.71** | 33.9 | 0.0 | **0.85** | 4.03 ± 0.09 | 3.88 ± 0.08 | 3.98 ± 0.09 | 3.89 ± 0.12 |
| *with 16min segments instead* | 3.83 | 0.70 | 46.1 | 0.0 | **0.85** | 3.83 ± 0.08 | 3.89 ± 0.08 | 3.87 ± 0.09 | 3.88 ± 0.09 |
| **SpeechSSM-9B** | **3.57** | **0.71** | **75.0** | 0.0 | **0.85** | 3.86 ± 0.08 | 3.90 ± 0.08 | 3.88 ± 0.07 | 3.95 ± 0.07 |
| *Ground Truth* | *3.61* | *1.00* | *100.0* | *50.0* | *0.92* | *4.12 ± 0.08* | *4.11 ± 0.08* | *4.10 ± 0.08* | *4.09 ± 0.08* |

*Table 4.* 16min completions of 10s prefixes of our LibriSpeech-Long test-clean. As there are no 16min ground truths, we only take reference-free metrics. ⊞ denotes model extension via slide-and-prompt. Win%$_\text{SSM-2B}$ denotes win rate over SpeechSSM-2B.

| Method | PPL↓ | Win%$_\text{SSM-2B}$↑ |
|---|---|---|
| GSLM (0.2B)⊞ | 4.48 | 18.4 |
| TWIST-1.3B⊞ | 4.76 | 17.5 |
| TWIST-7B⊞ | 4.45 | 32.8 |
| Spirit LM Expressive (7B)⊞ | 5.66 | 37.7 |
| **SpeechSSM-2B** | 3.59 | *50.0* |
| *with Transformer instead* | 5.91 | 24.9 |
| *with 16min segments instead* | 3.55 | 48.4 |
| **SpeechSSM-9B** | **3.39** | **68.3** |

ence, logicality and interestingness become increasingly apparent, showing that **side-by-side comparison versus LibriSpeech-Long ground-truths is a new and difficult benchmark** for (long-form) spoken language generation.

### 7.2. Proposed Extrapolation Metrics and Results

**Naturalness Mean Opinion Score over Time (N-MOS-$T$).** To balance cost and informativeness, for each example we select a 5 sec. span from each minute; specifically $[t_\text{prompt}, 60)$, $[60, 120)$, $[120, 180)$, and $[180, t_\text{max})$ where $t_\text{prompt}$ is prompt duration and $t_\text{max}$ is ground truth duration, and extract each span's audio from every model's generated continuation.

**Semantic Coherence over Length (SC-$L$).** To evaluate semantic faithfulness to the original prompt over time while mitigating the effect of speech rate, we take each continuation's transcripts and divide them into spans of 100 words (as determined by whitespace). Each 100-word segment represents ∼30 seconds of speech, with the advantage of normalizing out differences in generated speech rate as well as degenerate silences. Our SC scores are cosine similarities between the embedding of the original prompt $\mathbf{e}_\text{prompt}$ with

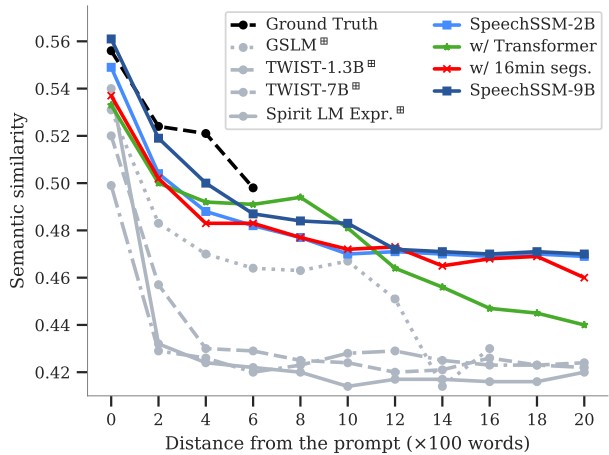

*Figure 5.* Semantic similarity between a 10s prompt and the 100-word segment starting at word $100L$ in the 16min completion, as measured by cosine similarity of Gecko embeddings (SC-$L$).

that of each 100-word segment $\mathbf{e}_{100\ell:100(\ell+1)}$.

These metrics, motivated in Section 4 and shown in Tables 3 and 4, show that **windowed model extension is a poor length extrapolator**. GSLM and TWIST are already low in our proposed N-MOS-$T$ by minute one, having trained on even shorter sequences. Spirit LM, which has seen sequences up to 200s (though rarely, due to text interleaves), degrades acoustically over time, though slower.

Our proposed SC-$L$ is plotted for 16 minutes in Figure 5 (table in Appendix A). As generation progresses, we see a decline in SC-$L$ scores, aligning with the natural flow of speech starting on a topic and evolving over time. However, existing models except GSLM sharply drop in semantic coherence (SC) scores around 200 words (∼1min; around their training lengths). GSLM fares better as its failures are untranscribed noise which do not show in SC-$L$ (but do

in N-MOS-$T$), but still worse than our models which are closest to the 3-4min ground truths' performance. At 16min (Table 4, Figure 5), we do not see obvious degradation from SpeechSSM-2B generating beyond its training length. In contrast, SpeechTransformer-2B had comparable PPL, win-rate, and SC scores up to its training length, but these metrics quickly degrade at 16min. This **suggests an edge for SSMs in long-form speech generation**, complementing past work (Gu et al., 2022). In all, SpeechSSM's design and architecture lead to generative length extrapolation.

### 7.3. Qualitative Discussion

We invite the reader to listen to the samples on our website. For convenience, Figure 2 shows that even SpeechSSM-2B generates intelligible and more coherent speech over the 4min duration, with ongoing references to the "Philip" in the prompt, along with new recurring entities like "Prince Maria", "Prince Albert", and "Horace Barrows" also consistently appear, maintaining contextual relevance. We provide further transcripts for ours as well as other models in Appendix E, showing how their decline in SC-$L$ is qualitatively visible as e.g., degeneration into repetitive outputs.

Given our use of windows during synthesis (Section 3), one may wonder if audio quality differs at transitions (every $25 + 23n$ seconds; the multiple comes from 30s, minus 3s for the prompt and 2s truncation per side). We ran human evaluation comparing 5-second windows centered at chunk boundaries vs. chunk midpoints, and found no clear effect on MOSes or six categories of rater-flaggable issues (Appendix A.2), which matches our informal listening.[7]

### 7.4. Inference Efficiency

**Throughput** measures the maximum number of tokens that can be successfully decoded per second given fixed memory, e.g., by increasing the batch size. In Figure 6, SpeechSSM attains higher throughput due to its recurrent layers that maintain a constant-size state; furthermore, its self-attention has a maximum context of size 2048, giving a lower bound. In all, this confirms SpeechSSM's capped memory use and thus capacity for unbounded generation, with >120x the throughput of SpeechTransformer once decoding 16.4k-token sequences in batch (also depicted in Appendix A.3). **Decoding time** measures the speed at which a sample can be decoded to a desired length Figure 7. We see that SpeechSSM also increases speed per step versus SpeechTransformer; an additional factor in SpeechSSM's improved throughput. On the TPU v5e, the 2B SpeechSSM decodes 16.4k tokens (roughly 10.9 minutes) in just over 100 seconds, a real-time factor well under 0.2x.

---

[7]That said, targeted listening does reveal e.g., subtle loudness shifts at times; still a massive reduction versus synthesizing without overlaps or without a fixed prompt.

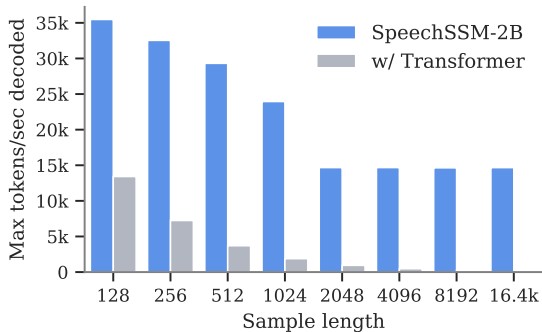

*Figure 6.* Max throughput under batch decoding per model and sampling length on one TPU v5e (unconditional generation).

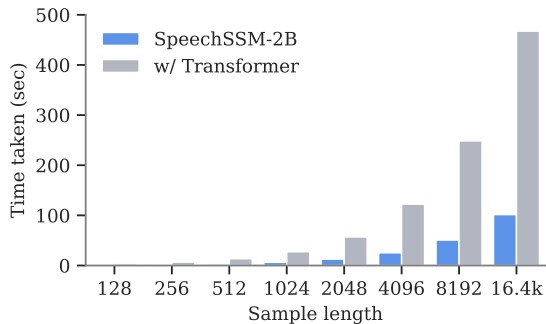

*Figure 7.* Time taken to decode a single sample (batch size 1) to a target length on one TPU v5e (unconditional generation).

### 7.5. Extemporaneous Speech Generation

With few exceptions (dGSLM, Spirit LM Expressive), most spoken LMs are trained on read speech such as Libri-Light. However, long-form multimedia and assistant applications likely require modeling of spontaneous speech, which has its own long-term discourse structures (e.g., podcasts; Nishimura et al., 2025). Hence, we also developed **SpeechSSM-X**, a 2B SpeechSSM model trained on a 216k-hour corpus of eXtemporaneous monologues (see Appendix B.4 for details). We find that it is able to generate natural monologue speech in a more informal, extemporaneous style, while showing similar coherence at the multi-sentence level; see our website for examples.

## 8. Conclusion

We considered the task of generative modeling of long-form speech. For modeling, this led us to SpeechSSM, the first spoken state-space LM, allowing generation than can go indefinitely without running out of memory. For evaluation we created the LibriSpeech-Long benchmark and proposed new evaluations for long-form speech continuation. We hope our work will simplify and enable new audio generation applications involving long-form media, such as audiobooks, podcasts, voice agent sessions, and video-related content.

## Acknowledgements

We are grateful to Zalán Borsos for their technical feedback; to Tongzhou Chen, Roshan Sharma, and members of our Speech and Griffin teams for infrastructure support; and to Soroosh Mariooryad for general assistance. We also thank the anonymous reviewers and area chair for their support and helpful feedback.

## Impact Statement

Progress on coherent, efficient, and unbounded audio-native speech language models will improve the availability and accessibility of audio-based human-computer interfaces, particularly towards the generation of speech with paralinguistic nuances often inexpressible in a standard text-to-speech prompt, as well as enabling models in primarily oral languages (textless NLP). It should also stimulate the use of synthetic speech in long-form creative multimedia applications, and could conceivably generalize to non-speech audio such as music.

We recognize that increased efficiencies in speech language modeling may increase the proliferation of audio deepfakes or low-quality synthetic media. However, for these uses we do not believe our work exacerbates what is already achievable (at higher content quality) by cascading text large language models (LLMs) with modern voice-promptable text-to-speech systems. We hope our work increases public awareness that direct methods in deep learning now enable large-scale generation of expressive speech, analogous to the situation in text media due to LLMs.

As in prior work for generative speech language modeling, speaker prompts used in our read-speech demos come from public domain LibriVox audiobooks and are only used to generate spoken continuations in the same setting, albeit for longer durations. For our spontaneous-speech demos, the voices have been explicitly licensed for speech synthesis. We do not currently release model weights.

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

# A. Additional Results

## A.1. SC-$L$ (16 min.)

*Table 5.* Semantic coherence scores are over the indicated spans of words for the transcripts of 16-minute completions. ⊞ denotes model extension via windowed generation. GSLM's blank is due to not generating that many words in the time period. Ground Truth's blanks are due to its audio being ≤4min.

| Method | SC-$L$↑ | | | |
|---|---|---|---|---|
| | 0-100 | 600-700 | 1200-1300 | 1800-1900 |
| GSLM (0.2B)⊞ | 0.531 | 0.464 | 0.451 | – |
| TWIST-1.3B⊞ | 0.499 | 0.420 | 0.429 | 0.423 |
| TWIST-7B⊞ | 0.520 | 0.429 | 0.420 | 0.423 |
| Spirit LM Expressive (7B)⊞ | 0.540 | 0.422 | 0.417 | 0.416 |
| **SpeechSSM-2B** | 0.549 | 0.482 | 0.471 | 0.470 |
| *with Transformer instead* | 0.533 | **0.491** | 0.464 | 0.445 |
| *with 16min segments instead* | 0.537 | 0.483 | **0.473** | 0.469 |
| **SpeechSSM-9B** | **0.561** | 0.487 | 0.472 | **0.471** |
| *Ground Truth* | *0.556* | *0.498* | *–* | *–* |

## A.2. Human Ratings around Synthesis Boundaries

From SpeechSSM-2B's long-form continuations to 4 minutes, we take our subset of 50 prompts (Appendix B.3). For each, we take five-second windows centered at synthesis chunk boundaries versus synthesis chunk centers, giving 10 windows of each type. Each gets 2 ratings, to give 1,000 ratings per condition (Table 6). The mean opinion scores (MOSes) are nearly identical (4.05±0.07 vs. 4.07±0.07) and there is no clear trend in synthesis issues, suggesting concatenation boundaries do not lead to net loss in naturalness to an attentive but non-specialist listener.

*Table 6.* Human evaluation of generation quality of SpeechSSM-2B at and away from chunk boundaries. We report Mean Opinion Scores (MOS) with 99% confidence intervals along with rater-flagged error categories.

| Location | MOS | Artifacts | Pronunciation | Speed | Prosody | Sentiment | Other |
|---|---|---|---|---|---|---|---|
| Chunk boundary | 4.05 ± 0.07 | 115 | 16 | 41 | 72 | 38 | 26 |
| Chunk center | 4.07 ± 0.07 | 122 | 16 | 53 | 78 | 32 | 20 |

## A.3. Relative Throughputs

In Figure 8 we compare SpeechSSM-2B's throughput versus SpeechTransformer.

# B. Additional Implementation Details

## B.1. Model Training and Selection

SpeechSSM-2B w/ 30s, 4min, and 16min segments get 750, 5760, and 24k tokens per segment respectively, under USM-v2's frame rate of 25Hz. For the 4min's length extrapolation (Section 3), we drop the last 10s of each segment.

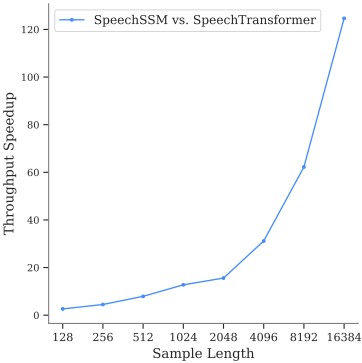

*Figure 8.* Ratio of SpeechSSM to SpeechTransformer's throughput on a single TPU v5e performing unconditional generation at different sampling lengths, based on Figure 6.

Each version is trained on groups of 512, 128, and 32 sequences respectively, amounting to 768k tokens per batch. We train with the Adam optimizer with a learning rate of 5e-4 and weight decay 0.6 for 100k steps (with a warmup of 1k steps and a cosine decay schedule to 1/20th the learning rate). We select the best model on a fixed random subset of LibriSpeech-Long dev-clean, by generating continuations with each model to its target length over 5 checkpoints, then choosing the one which minimizes transcript perplexity (Section 6.1).

### B.2. SoundStorm and SoundStream

Our SoundStorm follows the original hyperparameters for a voice-promptable model as described in Borsos et al. (2023b) (30s windows), except we double the number of layers to 24 to give 600M parameters, and our model was pretrained on a mixture of Common Voice v11.0 (Ardila et al., 2020), Multilingual LibriSpeech (Pratap et al., 2020), and VoxPopuli (Wang et al., 2021). This is comparable to other works which do not restrict their synthesizer data to LibriLight: TWIST (Hassid et al., 2023) does not explicitly indicate its vocoder data, but for HuBERT and the speech LM itself the English splits of these three datasets were used in training among many others; Spirit LM (Nguyen et al., 2025) used Expresso (Nguyen et al., 2023a) to train its vocoder.

### B.3. Model Evaluation

For text-based evaluations, we transcribe the generated speech to enable the application of natural language generation (NLG) metrics (Lakhotia et al., 2021). Unless stated otherwise, we use wav2vec2-base-960h (Baevski et al., 2020) for ASR, applied on 180s windows. To reduce cost and to mitigate length/duration as an indicator for ground truth, for more intensive evaluations (N-MOS, side-by-sides) we randomly selected 200 test-clean utterances with ground-truth continuations $\geq$ 7s.

**For long-form evaluations:** Gecko is a text embedding model trained on a variety of tasks including document retrieval, semantic similarity, and classification using long passages, and is more suited for extracting semantic embedding of long texts in open-domain generation. The choice of task is given via prompting; we use 'search results' as the prompt which was used to train on clustering tasks. For win-rates, to mitigate length bias we only consider the 193 examples $\geq$3.5min (71% of test-clean). For MOS computations, we randomly selected 50 from these.

### B.4. SpeechSSM-X

We use speaker-based detection to categorize audio files into long-form monologue content, with turns marked by voice activity detection. We take contiguous turns of up to 16min in length as training sequences.

## C. Additional MOS Evaluation Details

For short-form N-MOS, we took our subset of 200 prompts (Appendix B.3) and collected 6 ratings for each model continuation. For N-MOS-$T$, we took our subset of 50 prompts and collected 24 ratings for each model continuation and slice. Hence, each score represents 1200 ratings unless stated otherwise.

The following prompt was used in all cases:

```
This task requires you to listen to
audio clips using headphones in a quiet
 environment.

In this task, you will be given audio
clips. For each clip, please listen to
the speech very carefully and then
select a rating on a scale of 1 (very
unnatural) to 5 (very natural) using
0.5 point increments. The rating should
 be based on how natural or unnatural
the speech sounded. Please do not judge
 the grammar or the content of the
sentence. Instead, just focus on how
natural the speech sounds.

Possible Ratings
1: Bad
1.5
2: Poor
2.5
3: Fair
3.5
4: Good
4.5
5: Excellent
```

# D. LLM-as-a-Judge Example

This is our model prompt, with the example of comparing a transcript of the ground truth versus a transcript of GSLM's (Lakhotia et al., 2021) generation. This example was chosen to highlight the rater model's acknowledgement of the prompt request to not penalize incomplete sentences:

```
# Instructions

Please act as an impartial judge and evaluate the quality of two texts which occur in
the context of a book. These texts are transcribed from audio recordings that were
truncated to a fixed duration. Your job is to consider the following criteria to
evaluate which text is better:
- Fluency: How grammatically correct is the text?
- Coherence: How well do the sentences of the text fit together?
- Logicality: How much does the text obey common sense?
- Interestingness: How enjoyable is the text to read?

First, read text A and consider its fluency, coherence, logicality, and
interestingness. Do not penalize the text for ending mid-sentence or mid-paragraph.

Then, read text B and consider its fluency, coherence, logicality, and interestingness.
 Do not penalize the text for ending mid-sentence or mid-paragraph.

Afterwards, compare the fluency, coherence, logicality, and interestingness of the two
 texts. Do not penalize either text for ending mid-sentence or mid-paragraph.

Finally, after providing your explanations, you must output only one of the following
choices as your final verdict with a label:
1. Text A is significantly better: [[A>>B]]
2. Text A is slightly better: [[A>B]]
3. Tie, relatively the same: [[A=B]]
4. Text B is slightly better: [[B>A]]
5. Text B is significantly better: [[B>>A]]

Example output: "My final verdict is tie: [[A=B]]".

# Comparison task

## ---------- Text A ----------

Pearl accordingly ran to the bow window at the further end of the hall and looked
along the vista of a garden walk carpeted with closely shaven grass and bordered with
some

## ---------- Text B ----------

Pearl accordingly ran to the bow window at the further end of the table. "Palmer to
Austin and two boys," said the next fore keeper.

## ---------- Detailed Comparison of Continuations ----------
```

Below is the rater model's response:

```
**Fluency:** Both texts demonstrate good grammatical structure. Text A is a complete
sentence, while Text B ends mid-sentence, but this isn't penalized as per the
instructions. Thus, they are roughly equal in fluency.

**Coherence:** Text A presents a seemingly disconnected statement. Who is Palmer? Why
are Austin and two boys being mentioned? Without further context, it lacks coherence.
Text B describes Pearl's action and the scene she observes, creating a more coherent
picture.
```

```
**Logicality:** Text A lacks logicality due to the lack of context. It's a
grammatically correct sentence, but the information presented doesn't make much sense
on its own. Text B describes a plausible scenario \u2013 someone running to a window
and looking at a garden.

**Interestingness:** Text B is more interesting. The imagery of the garden walk with
closely shaven grass is more evocative than the seemingly random information presented
 in Text A.

Overall, Text B is superior due to its better coherence, logicality, and
interestingness, despite both having good fluency.

My final verdict is: Text B is slightly better: [[B>A]]
```

## E. Generated Samples

Figure 9 presents transcripts of 4min and 16min speech continuations generated by SpeechSSM-2B, and Figure 10 compares 4min speech continuations generated by SpeechSSM-2B with those from other models. The recurring proper nouns are highlighted in consistent colors, demonstrating SpeechSSM's superior performance in semantic coherence and context maintenance over long generations. '4min' denotes the default SpeechSSM-2B and '16min' the variant trained on 16min segments.

**SpeechSSM 4min**

On Sunday afternoon Alexander remembered Miss Bourgoin's invitation and called at her apartment. He found it a delightful little place and he met charming people there.

For five minutes when Alexander came downstairs, talking as long as he had time to know, he said that Miss Bourgoin had been shown some pictures. He seemed to be quite satisfied. He was afraid to talk to Alexander, not much of him. He was afraid of no one but Alexander and on this afternoon's recreation he heard her chattering away in his dining room.

He was very anxious to get to a picture for short but the picture pleased him and helped him much. At the first glance he found it decidedly refined and ornamental. He was wearing a handsome, dark mustache autumn. And the impression he made on him was especially great at the beginning, a large strip of pink and blue.

*(more speech...)*

There was one little picture which Mr. Bourgoin tried to put in his way but when he had one, he really carried it with him. If that wasn't all the entertainment they'd have to wait, he mentioned, and began putting in into his pocketbook, which was written on huge black needles, and which was surrounded by a confusion of rounded tiles and someone had a large scattered notebook in it. It was written in a pair of rather large crinoline boots that appeared every minute and made a pleasant dream. Mademoiselle did not make more than half a picture. This was a great experience with some pictures containing Alexander's pocketbook.

*(more speech...)*

"What time is it?" I asked, looking up at the handsome disembodied sky.

"You know, Mr. Bourgoin calls for it."

Mr. Bourgoin came downstairs and everyone came to see us. I liked the picture, and I found Mr. Bourgoin in the same attitude with a family card in his hands and having a private trouble face before him.

"I hope so, Mademoiselle," I said. "I suppose it is a shame to bang with a horse when he has not got any food."

**SpeechSSM 16min**

From the evil seed of lust, all other deadly sins had sprung forth. Pride in himself and contempt of others, covetousness in using money for their subsequent life, memory of poverty kindling the anger of the judge. Hairy, staining clothes of brothers, rum slaving sisterly and dear children in a pitiless passion. The white spots and sober face and rough, hard-mannered features, but wealth of evidence to decide. This distress would not be terrible.

*(more speech...)*

"And so you give me a reason for it." Said Tobinos. "Why then should you choose a reason for it? For why could I not accept what I believed to be good?" The yeast answered with an effort, "Lord, what is this that I have not said? You are skillful hatchet. I am a very little one to you." He added, "Ah, how fine is your friend! You were not at all fit to walk alone like a fox. And I am going to sit with you here. I labor to have my yeast among you in a better place. It will make you ask."

*(more speech...)*

"I want you to understand." Said Tobinos. "You want to know whether you deserve what you have given me to believe. Would you still like to believe it? Lord, I want to know whether I can believe and love all and such a man. I was there thinking, Lord, what will not I do? Lord, what will not I do?" Thereupon, Gorab lifted himself lightly from the wizard's chest, coughed, took his foot out of the chest, and said, "Lord, I labor to give you life and strength."

*(more speech...)*

"Go straight up. Be thou one of the men in the world and bless your pride." Came from Gorab as he opened his eyes. The red king was really blinded in body. And his glances contracted with a fixed look. He felt the coffer coming and going fast. Gorab stepped towards him and shouted some new well-known word to him, and ordered the wizard to bound him in a bowl of bread into the cave. "Mumgulul, Samko, wake up." Said Tobinos.

*(more speech...)*

 And he approached the worker at the same instant. "What is the consequence?" He said. A short time later, the young man, his body bent, showing one leg above the knee. His hands dropped to his cheek, and he sat up in the depths of his berth. "I must follow," said Tobinos, "and send the strongest of the soldiers." It was not possible to pass like a flash, as it is generally understood for a stranger to be so lying on the hammer. "Yes, who cometh forward to disgrace us? The Brazilian? Yes, but thou hast sold thy goods to the Brazilian? No, but I have already bought the Brazilian." Said Tobinos. He went, however, not in good humor, but with a desperate desire to delude his enemy by calling out the old man,

*(more speech...)*

Her right foot, with its long brown fingers, her fair hair, the truth sprang from her eyes. "Arthur can't afford to do this." She said. "And it's no use scolding me or me alone. And besides, he might well feel the wrong way with his gun." She turned her face away, but still, he went on, "Let me look at the man who did those things."

*Figure 9.* Automated transcripts of 4min and 16min speech continuations generated by SpeechSSM-2B, atop a 10-second audio-only prompt from our proposed LibriSpeech-Long (test-clean). The prompt is underlined, and parts abbreviated with *(more speech...)* for emphasis. Recurring proper nouns are highlighted in consistent colors to show SpeechSSM's relative semantic consistency over time.

**Prompt (ASR):** Yes, it must be confessed, Polly was homesick. All her imaginations of her mother's hard work, increased by her absence, loomed up before her till

**GSLM Continuation (ASR):**
she felt the weight of their last fraction. And then she paused, pausing upon the poor child, while George was waiting for her.
"What? What?" **[Noise]** What kind of a man could he be in his house if he's going to...? He turned his head and stepped across the lawn, looking down at the window, then turned to her with a grave smile—**[Noise]**—at which all the rest of that piece of sound came, and Bonnie's face was pale. His hair grew full and stately. His head began to brag and faint, and he breathed heavily, feeling quite certain that Polly could be in his house by digging bread for money for his poor child, and having bread for money, and having bread in his hand for Bonnie's.
"Oh, mother," cried Polly indignantly, "You ought to wait on Polly." **[Noise]** "That's me, too." There I lay as if she did not laugh and entreat.
How long, no one could tell. The magistrate suddenly assumed surprise, followed by a burst of delight, and the more today. Now, the milk and supper with those words of selfishness, she had written home better than she was before, **[Noise]**
*(more speech...)*
Thoughtful that the man **[Noise]**, who is wonderful, would not be the slave, but the master. He opposed his limbs to be upon the top with the worthy company. Accordingly,

**TWIST1.3B Continuation (ASR):**
teatime. The little boy was suddenly cold and told of his trouble. Two of the poor old black women whom she had visited for the purpose of looking at the scene, and contemplating the wretchedness of the crime. The purchase of a prison, the pardoning Babylon and apology. The incorruptible tidurage for toic with epicures, **[Noise]**
*(more speech...)*
However, the burger. Burger, burger, burger, burger, burger, burger, burger. Burger, burger, burger, burger, burger, burger, burger, burger, burger, burger, burger. Burger, burger, burger, burger, burger, burger, burger, burger, burger, burger, burger, burger, burger, burger.

**TWIST7B Continuation (ASR):**
she was ashamed to continue her studies, and he could not keep her thoughts from occupying herself with Abby, who was so devoted to her. He had put her in the way of enjoying herself when she was young, and now she was in the way of giving herself her willingly to others. He rang a hand bell at the door. "Mrs. Manstay, if you please," said the housekeeper in halting tones, giving Hannah to stand as he brought her to the door. "Where's Hannah Winnyard?" Mrs. Manstay went again to her window, and heard and saw, or felt she did in what the schoolmaster was doing. The boy was dressed in the dark blue gown and broad white collar that she might keep her sight of him. **[Noise]** Mrs. Manstay said nothing, putting her temp to him. **[Noise]** She had two families to attend to. She got out to catch the night train, but the meaning which the moment of felt gave her a slight shock, and she looked down at the piece of paper without a word. **[Noise]**
*(more speech...)*
So, one thing that I want to do, I want to know when you think of Star Trek, what do you think of? [Noise] Star Trek. You think about this. Okay. What do you think of Star Trek? You Star Trek. What do you think of? What do people **[Noise]** That would be Star Trek. I mean, that would be Star That would be Star That would be Star Trek and then you're like "Isn't Star Trek a trend?" **[Noise]** like medieval of Star Trek. No, it's not a trend.**[Noise]**

**Spirit LM Continuation (ASR):**
they seemed a reality. She was a trifle prettier than usual, but she is not a child of one any longer. And there was no optimism here for the paid-to hope that she would be able to keep Java completely supplemented by the family happiness. At the time Lucy was a child of 14. She might taken make that worried to the village public school to Caesar, the rotations of the bullion had been completed, she could kind of hide the math global music time in public education for a little while perhaps and then go sit down amongst successful students. But it would be better not to try to earn the money to pay some students by compromising discriminate traders of under the provisions of the United States Constitution. Thank you. Coming up. Thank you, your honor. Think you satisfy that request as relates to tax compilers promotion amount of money damages, that crossed the line between constitutional and state forums.
*(more speech...)*
There is also different impression on the record which are entirely distinct from this case. There was no suppression, but no distraction in the telling X plus the member of suspicion him committing taking down turning him back to the carriage and looking for him in order to get rid of him which was wanting to take him something closer to the Larson but she wouldn't to going back that he's close to death and he couldn't wait for him so he went back to get the arm chair.
*(more speech...)*
That's why I feel like a high school student. I feel like I'm in a 1960 student. That's why I can't go back to secure me. That's why I can't be put in, they can be stealing coffins, I can't put that in the end of comparisons. That's why I can't be put a field of leadership and comparison. That's why I can be a teacher so cares. That's why dad said I can't be, that's why I can't be practical. That's why I can't be see.

**SpeechTransformer Continuation (ASR):**
she hardly remembered them nor anyone else. She was, for a boarding girl, 18 years old, and any man who had some influence over her that was to be would find it out. Your kennel men, you see, she was for a choral business. I do believe you would call more work on a professional matter, a day or two, young ladies, with you at nights to catch sight of. She's that grown up, isn't she?" "Oh, it's a pity, uncle," Annie exploded with feeling. There's six boys, and the lads and one, I don't know which. The widow here has a little girl of her own, Geoffrey and Chuffy, before whom you two meet on Shendent Bullen platforms. "Allow me to select for you a girl." said Polly, and Dolly came. Her mother raised her up springless, and Billy marshaled his friends in thick garments. She had to go down to his office next and take her place at his hotel. The boarder of Grace's offices came in, all smiles and greetings, and Dolly made a call.
*(more speech...)*
He brooded over her until he contemplated his new dress. "Don't you look as homely as all that, Bobby dear?" murmured Pinocchio solemnly. "Oh, don't I wear?" asserted Annie. And smiled brightly after the little Gustave, who was half menacing in his goat. His face grew very pale, and his rumpled brown hair fell in drags in front of him. Held up in benignant readiness as a long tunic of just that shape she had left. "There's something telling, Mr. Cart," began Dolly, looking silently down. "Nothing, I'd rather the overseer do it in this name."
*(more speech...)*
How many more came out? Three chains and old stuff. "Where?" demanded Polly. Three for Miss Gould, who had silver beads. I'm sure she didn't wear any of her old beads. Annie gathered herself together, and patting her adoring head back at the doorstep, strutted away over the clearing. Polly lowered her old, gentlemanly head. I didn't have time to go over to Lady's Place. Then Miss Gibson is up. And I'm sure she wants.

**SpeechSSM Continuation (ASR):**
she had fallen into an ugly sleep. When she awoke, the children were sleeping, but not frightened at all, for fear they should stir a step, and one had to wake them before she awoke. And then she heard a gust snap and fall from her bedroom window. Mrs. Hopkins sprang from her tray and dressed herself and her mother, and then ran to wash her face, but she was still busy, and brought up tea. "Who is Polly?" asked Mr. Hampton as she laid down her tea. "Polly? She hasn't eaten anything already!" The children, overcome with grief, turned to leave the room. And for some moments the girl shivered and wept because, when she awakened, the room smelled of bad, cold milk. That night, they were more worried than they had expected. They found the handcuffs driven in the kitchen silk, and the pot hammer still burning in the kitchen fireplace. They'd said when and how they were to come. The only thing that startled Polly and her mother was the notion of the handcuffs being driven in Mr. Hampton's
*(more speech...)*
I'm different from Polly and I want her to be a hero too. She's a very nasty child and doesn't know exactly how to speak to me." "That's the kind of life the old woman, the poor old woman, goes into now. I must not give up my mother. It's right to have a continuation in still things." "Don't you think she's good enough for me?" asked Elizabeth spoken slowly. "Do you?" "Not any of us. This is too horrible for delay. I couldn't draw around you and tell you to lie down as well as I can. When perhaps, I can tell you that little hand of yours after you've been sleeping, and get away all the same." "Why are you going away so quietly?" "I can't remain, Polly."
*(more speech...)*
You must not abandon the idea, Elizabeth, the flesh is too weak, it's too stealthy and it hurts. It is horrible to say so. "That I don't know, it isn't worth living. If you have no sympathy with my dislike, you shall have more too." He stood by the door with a brave, small voice as he wiped his dry face and patted her on the arm.

*Figure 10.* Automated transcripts of 4min speech continuations generated by SpeechSSM and baselines. Parts abbreviated with *(more speech...)* for emphasis. Recurring proper nouns are highlighted in consistent colors to show SpeechSSM's relative semantic consistency over time. Non-sense sentences and proper noun errors are highlighted in grey.

