# OpenReview forum: "Long-Form Speech Generation with Spoken Language Models"
_ICML.cc/2025/Conference — ICML 2025 oral_

### Official Review · Reviewer_QbKA · 2025-03-03

**Overall Recommendation:** 3

**Summary:**

This paper aims to develop a speech language model (LM) capable of modeling long-form speech. The key distinctions of this paper compared to previous studies are as follows: (1) Introducing State Space Models (SSMs) into the speech LM at the speech-token level, resulting in improved perplexity (PPL) and other evaluation metrics compared to existing baselines. (2) Highlighting shortcomings in previous evaluation methods for speech LMs, proposing a more effective evaluation approach, and introducing a new benchmark, "LibriSpeech-Long," specifically designed to evaluate these capabilities.

## update after rebuttal
As I mentioned in my official comment below, I have read the authors' rebuttal and slightly increased my score accordingly. Once again, thank you for your prompt and thoughtful response.

**Claims And Evidence:**

The authors have established several criteria that a speech LM designed to model long-form speech should satisfy. They also described how State Space Models (SSM), along with their proposed model and dataset, were employed to meet these criteria.

**Essential References Not Discussed:**

I found no issues with references.

**Experimental Designs Or Analyses:**

I have some questions specifically about the evaluation rather than the experimental design itself. I'll address these points separately below.

**Methods And Evaluation Criteria:**

I have a few questions below regarding the evaluation scheme, specifically parts that were slightly unclear or raised my curiosity. Aside from these points, everything else seems appropriate.

**Other Comments Or Suggestions:**

The paper is well-written, and I did not notice obvious typos.

**Other Strengths And Weaknesses:**

### Strengths
1. Introducing SSM at the speech-token level to enable effective long-form modeling.
2. Proposing both a benchmark and novel metrics designed explicitly for evaluating long-form speech generation.
3. Overall, the approach is reasonable, and the authors have leveraged the advantages of SSM for long-form generation.

### Weaknesses
1. Regarding the N-MOS (ignoring grammar and content) and SpkrSim metrics, I believe these primarily reflect the qualities inherited from speech tokens proposed by previous works such as USM-v2 or SoundStream, and the speech decoder (SoundStorm), rather than being novel contributions of this paper.
- If the goal was to compare these aspects, the baselines should also have been trained as personalized speech synthesis models, or similarly, the USM-v2 tokens used by baselines should have been decoded with a separately trained vocoder such as HiFi-GAN, similar to the previous works. (The controlled variables should have been introduced only in the proposed SSM-based model.)

2. Regarding the perplexity (PPL) results, the authors show that SSM is effective compared to Transformers. However, other metric, particularly for the N-MOS results in Tables 2 and especially 3, I notice the confidence intervals are notably large. I am curious why these confidence intervals are so wide, what the corresponding p-values are, and whether it is indeed possible to conclusively determine the advantage of SSM over Transformer-based models based on these numbers.

3. In terms of novelty, I personally feel the contributions might be somewhat limited. Aside from introducing a new benchmark and proposing new evaluation metrics, the main impression is that the paper simply applied SSM to speech tokens. If I have missed any additional structural modifications or specialized adaptations specifically designed for speech data within the proposed model, I would appreciate further clarification from the authors.

**Questions For Authors:**

1. In recent language model trends, scalability has emerged as an important factor. I am curious about how the performance of the proposed SSM-based approach changes as the model scales up in Speech LLMs. Specifically, would performance significantly improve with increased scale, or is there a clear performance upper bound?

2. Additionally, during speech generation, the authors mention generating audio chunks with overlapping segments for memory efficiency, then concatenating them. I would like to know whether this concatenation process introduces any artifacts at the boundaries. If artifacts exist, it would be helpful to understand their nature.

**Relation To Broader Scientific Literature:**

I agree with the authors' contribution in integrating SSM into a token-level speech LM, which has not been addressed by previous studies. Additionally, highlighting the limitations of existing evaluation methods and proposing an improved evaluation metric also constitute contributions of this paper.

**Theoretical Claims:**

Since this paper focuses primarily on applying SSM, there do not appear to be specific issues regarding proofs or theoretical claims.

---

> ### Author Rebuttal · Authors · 2025-04-01
>
> Thank you for your thorough reading of our work!
>
> > W1: SpkrSim and N-MOS reflects speech tokens; decoupled tokenizer/vocoder baselines?
>
> **We agree N-MOS and SpkrSim are primarily due to tokenizer/vocoder**, hence why it felt unnecessary to isolate them from the backbone architecture change as these metrics would focus on our "acoustic" improvements. Then, to isolate backbone improvements from tokenizer/vocoder, we introduced SpeechTransformer as a baseline, sharing the same speech tokenizer (USM-v2) and vocoder (SoundStorm) with SpeechSSM.
>
> Also, despite using existing tokens, **we view our _deliberate integration_ as the contribution**. Other schemes entangle acoustic and semantic properties, and so their N-MOS is tied to long-form ability (-T); ours is not due to our fixed context and semantic-to-acoustic stage. Other models have speaker drift, while our speaker-invariant tokens (USMv2) *and* speaker-prompted acoustic stage ensure even and high SpkrSim.
>
> > W2: N-MOS sufficient for SSM over Transformer?
>
> Due to shared tokenizer/vocoder/etc., **the N-MOS gap being narrow between SpeechTransformer and SpeechSSM is _expected_**. This is also why our proposed metrics are important, such as Win%, PPLs at 4x train (Tab. 4) and SC-L (Fig 5), isolating the semantic delta of SSM vs. Transformer.
>
> Regardless, **we’ve greatly expanded our human evals. We report(ed) 99% CIs**, where each model had 200 items, 3 ratings each (incorrectly ‘5’ in Appendix C). We 2x this to get **[tighter intervals for Table 2](https://raw.githubusercontent.com/demo474/ID4403_Demo/refs/heads/master/figures/rebuttal-updated-table2.png)**.
>
> Our N-MOS-T was a different, smaller rater pool due to technical/time issues. We've now aligned it with short-form N-MOS to give **[far better 99% intervals for Table 3](https://raw.githubusercontent.com/demo474/ID4403_Demo/refs/heads/master/figures/rebuttal-updated-table3.PNG)**: 1200 ratings per (model, slice).
>
> While magnitudes in the top range decreased, trends remain. In particular, **SpeechSSM now clearly improves over SpeechTransformer in N-MOS-T**; their 99% intervals do not even overlap in two of the columns. This also **shows N-MOS-T is affected by the backbone model also.**
>
> >  W3: Simply applied SSM; modifications/adaptations specifically designed for speech data?
>
> As our primary contribution is advancing long-form speech generation over tens of minutes, and being the first work on the topic, we outlined the task's minimum requirements (Section 3, first paragraph) and **intentionally meet them in a straightforward way**. Our architecture, benchmark, and evals _stem from the task_ -- issues with the obvious SpeechTransformer baseline, issues with metrics that capture acoustic issues rather than semantic quality, etc.
>
> This did involve key modifications: using hybrid SSMs from text LMs; disentangling speaker identity, acoustic, and semantic aspects; windowed tokenizing and decoding; repeated padding and avoiding implicit EOSes; using NoPE. Though these are not architectural changes in the narrow sense, they contribute to length generalization, enabling the model to effectively generate speech of unbounded length.
>
> >  Q1: Would performance significantly improve with increased scale?
>
> We expect performance to continue improving beyond 2B. This expectation is supported by the Griffin paper, the underlying hybrid recurrent architecture used by RecurrentGemma and thus SpeechSSM, which showed consistent gains across 1B, 3B, 7B, and 14B models on downstream text tasks.
> Towards this, **we have started training a SpeechSSM 9B**, initialized from RecurrentGemma 9B which was unavailable at the start of our work. **[Its training curve versus 2B is very promising and suggests headroom](https://raw.githubusercontent.com/demo474/ID4403_Demo/refs/heads/master/figures/rebuttal-9b-inprogress.png).**
>
> > Q2: Would concatenation process introduce artifacts at the boundaries?
>
> Yes, **there are artifacts due to waveform splicing, but they are very subtle due to overlapping,** as the mismatch in the contexts used to generate audio before vs. after the splice only starts 50 tokens (2s) away.
>
> Since SoundStorm uses 3s for speaker prompting, it synthesizes in 27s chunks, whose last 4s are overlapped with the next chunk. Hence, **when boundary artifacts occur, they occur at `25 + 23*N` seconds.** In the third file on the [demo page](https://demo474.github.io/ID4403_Demo/), at :25, the male voice gets briefly louder.
>
> We take 5s windows centered at the boundaries (“at concat”) and 5s windows at the chunk centers (“not at concat”). There are 10 of each window type in a 240s generation, giving 50 x 10 x 2 ratings/item = 1000 ratings per type. At 99% CI we get 4.05 ± 0.07 vs. 4.07 ± 0.07, showing **artifacts do not appear in MOS, [nor do they appear systematically in the categories raters can flag](https://raw.githubusercontent.com/demo474/ID4403_Demo/refs/heads/master/figures/rebuttal-concat-artifacts.PNG).**

---

> > ### Comment · Reviewer_QbKA · 2025-04-02
> >
> > Thank you for the kind response. The authors have addressed all the concerns I raised. Therefore, I will slightly increase my score.

---

### Official Review · Reviewer_qRRp · 2025-03-12

**Overall Recommendation:** 4

**Summary:**

This paper introduces the first speech-language model ***SpeechSSM***. Two new metrics and a new benchmark are proposed. Experiments and analysis are comprehensive.

**Claims And Evidence:**

**Yes**

**Essential References Not Discussed:**

**Maybe not**

I thought the related works were discussed.

**Experimental Designs Or Analyses:**

**Yes**

The authors conduct experiments for short- and long-form speech generation separately.

The experiments are well-designed. Comparisons and evaluations are comprehensive.

**Methods And Evaluation Criteria:**

**Yes**

**Other Comments Or Suggestions:**

There is no more comments from the reviewer.

**Other Strengths And Weaknesses:**

**Strengths:**
1. This paper is well-written. The figures and tables are clear and easy to understand.
2. The results generated by the proposed method are very competitive.
3. As the authors claimed, the proposed methods significantly increase the audio length during inference and training, which are meaningful for real-world applications.
4. The two new metrics and the new benchmark designed for long-form speech are useful.


**Weaknesses:**
1. (Minor) The architecture of SpeechSSM appears somewhat straightforward and could benefit from more innovative design considerations.
2. (Minor) The windowed tokenization and decoding approach represents a compromise solution rather than an optimal one.

**Questions For Authors:**

The reviewer acknowledges that the weights are currently not released. Will the model and weights be released in the future? This is important for further research.

**Relation To Broader Scientific Literature:**

The key contributions of this paper include:
1. **SpeechSSM** -- the first speech-language model for long-form speech.
2. Two new metrics and a new benchmark for long-form speech evaluation.

The related works seem to be discussed in the Section 2.

**Theoretical Claims:**

**Yes**

There is no theoretical claim in this paper.

---

> ### Author Rebuttal · Authors · 2025-04-01
>
> Thank you for your review and support!
>
> > (Minor) The architecture of SpeechSSM appears somewhat straightforward and could benefit from more innovative design considerations.
>
> As our primary contribution is advancing long-form speech generation over tens of minutes, and being the first work on the topic, we outlined the task's minimum requirements (Section 3, first paragraph) and **intentionally meet them in a straightforward way** so that we could also focus on dataset and metric design. Nonetheless, this already involved several key design choices, such as: using hybrid SSMs from text LMs; disentangling speaker identity, acoustic, and semantic aspects; windowed tokenizing and decoding; repeated padding and avoiding implicit EOSes; using NoPE.
>
> By establishing a strong baseline and benchmark, we hope future work can focus on new and innovative architectures for long-form speech.
>
> > (Minor) The windowed tokenization and decoding approach represents a compromise solution rather than an optimal one.
>
> The overlap windowing approach was a practical solution to effectively utilize a bounded tokenizer and decoder to achieve unbounded speech generation. Future work could explore streaming tokenizers to handle extended speech generation, but these have to train from scratch.
>
> In contrast, our method works on default non-causal tokenizers like HuBERT and USMv2. **It is close to optimal from the viewpoint of "working out of the box", and also from mitigating boundary artifacts**, as we quantify in the end of reply to Reviewer QbKA (#4). We consider our design and comprehensive details (e.g. avoiding implicit EOSes, where to speaker prompt) a contribution in its own right.
>
> > The reviewer acknowledges that the weights are currently not released. Will the model and weights be released in the future? This is important for further research.
>
> **SpeechSSM is largely RecurrentGemma finetuned on a public dataset** and so we believe it is easily reproduced. As USMv2 is not widely available, another speech tokenizer would be required, of which there are now many viable candidates (e.g., SpeechTokenizer, Mimi). Due to safety considerations and a proprietary dataset we are unlikely to release SpeechSSM-X.
>
> While for institutional reasons we can't promise a full release of the (LibriLight) SpeechSSM, **we could release finetuning code, e.g., atop [the Gemma finetuning library](https://gemma-llm.readthedocs.io/en/latest/colab_finetuning.html) and a public speech tokenizer**, to support further research.

---

> > ### Comment · Reviewer_qRRp · 2025-04-03
> >
> > The reivewer has read the rebuttal from the authors. All the concerns from the reviewer have been addressed.
> >
> > The reviewer decide to keep the final rating as 4 (accept).

---

### Official Review · Reviewer_9NCy · 2025-03-13

**Overall Recommendation:** 4

**Summary:**

The paper proposes SpeechSSM, a spoken language model designed for long-form speech generation. It is based on state-space models enabling efficient generation with constant memory consumption. To evaluate the model on long generations the authors propose LibriSpeech-Long benchmark and new evaluation metrics including embedding-based similarity, LLM-judged scoring, and time-stratified quality assessments. Results show that SpeechSSM performs on par with existing spoken language models for short-form generations and outperforms them on long-form speech generation.

**Claims And Evidence:**

The paper makes the following main claims:
  1. Model-wise: SpeechSSM is the first state-space language model for speech and provides long-form speech generation
  2. Evaluation-wise: The paper introduces new evaluation metrics for long-form speech generation, including side-by-side LLM-as-judge scoring and time-stratified quality assessments.

The paper provides clear and convincing evidence for both claims through experimental comparisons.

**Essential References Not Discussed:**

NA

**Experimental Designs Or Analyses:**

Two main experiments have been performed:

  1. Short-Form Continuation Experiments: The model is evaluated on 7s continuations given a 3s prompt assessing short-term coherence and speaker similarity.
  2. Long-Form Generation Experiments: The model generates up to 16 minutes of speech from a 10s prompt, evaluating semantic coherence, fluency, and speaker consistency across extended durations.

Main weaknesses:

1. No human evaluation is performed beyond N-MOS. While the authors employ LLM-as-a-judge, a human LLM correlation analysis would be very interesting.
2. Limited dataset diversity in training and evaluation. The proposed model is trained and evaluated on the same audio/speaker conditions (e.g. background/noise, neutral speech) whereas the other models have been trained on different data. If experiments on different datasets are provided it would demonstrate the robustness of your approach.

**Methods And Evaluation Criteria:**

The proposed methods and evaluation criteria in the paper are well-suited for the problem of long-form speech generation. To evaluate the model's performance the authors propose four evaluation methods:
1. LibriSpeech-Long benchmark: To assess the model’s performance on extended speech continuations the authors reformatted the original LibriSpeech dataset into longer 3-4 minute segments creating the LibriSpeech-Long benchmark.
2. Embedding-Based Metrics: Semantic embedding-based metrics are used to evaluate the content preservation of the generated speech.
3. LLM-as-Judge: Large language models are utilized as judges to provide side-by-side evaluations of generated speech samples.
4. Time-Stratified Evaluations: The paper introduces time-stratified quality assessments to analyze the consistency of speech over different time intervals within the long-form generation.

**Other Comments Or Suggestions:**

NA

**Other Strengths And Weaknesses:**

Strengths

1. Efficient and scalable approach. SpeechSSM has constant memory complexity and linear-time inference.
2. New evaluation metrics and the LibriSpeech-Long benchmark are important long-form speech research.

Weakness

1. Ablation analysis is missing. How different architectural choices (e.g. hyperparameters) impact model performance?
2. Only audiobook generations are considered. How would the model perform on other scenarios like dialogues?

**Questions For Authors:**

1. How does the choice of windowing parameters impact performance? How do different token widths and overlap sizes influence the model’s performance?
2. How does your model perform with different prompt durations? For example, why did you choose 10 seconds for long-form generation instead of 3 or 5 seconds?
3. Have you evaluated SpeechSSM on more expressive speech datasets, such as EXPRESSO or EmoV?
4. Have you tested the model for longer than 16min speech generation (e.g. 60 min)? Does it maintain semantic coherence and speaker identity?
5. How does your model handle non-linguistic vocalizations such as laughter or sighs?

**Relation To Broader Scientific Literature:**

The paper's key contributions align closely with advancements in the broader scientific literature. In particular, while state-space models have been used in speech generation, none have been applied as a spoken language model operating directly on acoustic tokens. Unlike prior transformer-based approaches such as TWIST and AudioLM, the proposed method leverages state-space models for more efficient long-form generation with constant memory complexity.

Additionally, the paper builds upon standard evaluation metrics for short-form generation and extends them for long-form evaluation. First, it introduces the LibriSpeech-Long dataset, created by reformatting the LibriSpeech dev and test sets to enable longer speech evaluations. To evaluate long-form generated samples it proposes new evaluation metrics for assessing the quality and coherence of extended speech generation (LLM-as-Judge, embedding-based metrics and time-stratified evaluation).

**Theoretical Claims:**

N/A

---

> ### Author Rebuttal · Authors · 2025-04-01
>
> Thank you for your review and thoughtful feedback about our work!
>
> > **Major Weaknesses:**
> >
> > No human evaluation beyond N-MOS
>
> Note **we have strengthened our MOS results**; see response to Reviewer QbKA (#4).
>
> > LLM-as-judge vs. human correlation analysis
>
> LLM judges align closely with human preferences (over 80% agreement) in pairwise comparison (Zheng et al, 2023), with SxSes more stable than single-answer grading. When we started, our LLM as judge (Gemini 1.5 Pro) was [the top generative LLM-as-judge](https://x.com/aseveryn/status/1793605627232833887) on [RewardBench](https://huggingface.co/spaces/allenai/reward-bench), with a score of **88% (roughly viewable as human agreement %)**. Our rubric is based on story generation evaluations (Section 5.2 + Appendix D), a reading/style comprehension task we believe is similar to tasks evaluated by these works.
>
> That said, a correlation analysis specific to our transcript SxS would be informative; we'll investigate this for the camera ready.
>
> > Limited dataset diversity... proposed model is trained and evaluated on the same audio/speaker conditions ... other models have been trained on different data. Experiments on different datasets...
> >
> > **Other Weaknesses:**
> >
> > Only audiobook generations... other scenarios like dialogues?
>
> We restricted our main model to LibriLight for fair comparison with non-Spirit LM works. However, to demonstrate diversity, **in 6.4 we also trained a version (SpeechSSM-X) on extemporaneous podcast-style monologue.** Our model **replicated this data's expressive and varied nature**, as can be heard in the [SpeechSSM-X samples on our demo page](https://demo474.github.io/ID4403_Demo/). That said, our models generate a single voice as they disentangle speaker identity (a design contribution of our work). **We see no blockers to training a model with entangled tokens to model many voices in dialogue.**
>
> > Ablation... architectural choices (e.g. hyperparameters) impact model performance?
>
> We ablated train/test lengths and Transformer vs. SSM. **We also now ablate text LM initialization and model size**; see Reviewer k44A (#1) and Reviewer QbKA (#4) rebuttals for details. We could explicitly discuss more things we tried e.g., RoPE length issues or overlap parameters in the camera ready.
>
> > **Q:** EXPRESSO, EmoV?
>
> To quantify SpeechSSM-X's performance, we are trying Expresso and could share results during this period if completed in time. We are considering context-based continuation (e.g. ["Subjective metrics" by Sesame](https://www.sesame.com/research/crossing_the_uncanny_valley_of_voice)) but welcome suggestions.
>
> > **Q:** window parameters, [tokenizer] overlap, and performance
>
> For our top concern of unbounded generation, we found that avoiding "implicit EOSes" was key (Section 3), so we used the largest window supported by USMv2; we only overlap to avoid edge tokens. It's possible that intentionally smaller windows may improve performance, which we leave to future work.
>
> > token widths and performance
>
> Longer audio token widths are an active research area which we believe is driven by poor long-form performance of current models. **We intentionally focus on architecture/training improvements _instead_ of larger tokens**, which we'll clarify in the camera ready. Future work can combine both approaches.
>
> > [synthesizer] overlap and performance
>
> We tested 0s, 2s, 4s, and 8s overlaps, and found that **no overlap produces artifacts at boundaries**. The smallest overlap we tried already significantly reduced these artifacts, so we chose it. See final part of our response to Reviewer QbKA (#4) for metrics.
>
> > **Q:** different prompt durations? 10 seconds for long-form generation instead of 3 or 5 seconds?
>
> **Shorter prompts are insufficient initial context to act as anchor** to compare against for long-term coherence, e.g., our SC-L metric of semantic similarity between the prompt and segments of the continuation. A prompt of at least 10 seconds (1-2 sentences) differentiates models (Figure 5) and constrains the space of valid continuations for LLM judging.
>
> > **Q:** >16min speech generation; semantic coherence and speaker identity?
>
> **We have generated with SpeechSSM up to 30 minutes** and see no issues going further. The model maintains speaker identity and speech naturalness indefinitely (as expected from moving speaker modeling to the acoustic layer), with a gradual semantic drift as one may expect from SC-L scores in Figure 5. However, paragraph+ coherence issues [seen in our 16min samples](https://demo474.github.io/ID4403_Demo/) should be improved first before such generations are worthwhile.
>
> > **Q:** non-linguistic vocalizations such as laughter or sighs?
>
> As long as the model is trained on data with such vocalizations, it can learn to reproduce them. **In the third [demo page](https://demo474.github.io/ID4403_Demo/) sample, from 45s - 1:15s, there are sighs, exhalation, and fillers (“uh”, “um”, “uh-huh”).**

---

### Official Review · Reviewer_k44A · 2025-03-14

**Overall Recommendation:** 4

**Summary:**

The paper proposes SpeechSSM, a spoken language model based on state-space models that can generate long-form audio in a single pass. The paper also proposes reference-based semantic similarity and LLM-based pairwise judgment to evaluate the generated long-form audio. They also released a new dataset LibriSpeech-Long for evaluating speech continuation. The SpeechSSM outperforms the baselines in both short-form and long-form generation. The new metrics and dataset enable better evaluation for long-form speech generation.

## update after rebuttal: I decided to keep my score.

**Claims And Evidence:**

The claims made in the submission are supported by clear and convincing evidence.

**Essential References Not Discussed:**

None

**Experimental Designs Or Analyses:**

The baselines are extensive and comparable.
The analysis on semantic coherence and error modes is comprehensive.
The Librispeech-Long benchmark is well-designed for the task.

**Methods And Evaluation Criteria:**

The proposed methods and evaluation criteria make sense for the problem at hand.

**Other Comments Or Suggestions:**

None

**Other Strengths And Weaknesses:**

None

**Questions For Authors:**

How important is the text-based initialization (using RecurrentGemma-2B) for speechSSM's performance?

**Relation To Broader Scientific Literature:**

The core contribution of this work is built on the developments in many different areas, most importantly language models and speech representation learning. It deals with the problem of spoken language modeling, with the core challenge being the choice of the right model and speech representation for modeling.
Previous approaches, like GSLM, used transformers to model speech tokens from HuBERT. Later, speech representation modeling improved. SoundStorm, for example, incorporated hierarchical modeling with RVQ which inspired AudioLM to do better modeling. However, the previous approaches were not good at modeling long speech sequences. That was until state space models like S4 arrived and substantially improved time-series modeling for long sequences.
For the evaluation methods introduced in this paper, it is also backed by the recent advancements in large language models (LLMs). Many papers are now starting to make use of LLMs to build evaluation metrics for their methods.

**Theoretical Claims:**

The paper does not have any proofs for theoretical claims.

---

> ### Author Rebuttal · Authors · 2025-04-01
>
> Thank you for your comprehensive reading of our work!
>
> > How important is the text-based initialization (using RecurrentGemma-2B) for speechSSM's performance?
>
> Following the insights from Hassid et al. (2023)'s TWIST model, which demonstrated that initializing spoken language models with pretrained textual knowledge enhances semantic coherence and fluency, we initialized SpeechSSM with RecurrentGemma 2B IT’s weights. As both SpeechTransformer and SpeechSSM are initialized with first-generation Gemma 2B models, we believe **the text-based initialization is not important to SpeechSSM’s gains over SpeechTransformer**. Furthermore, Spirit LM is initialized with Llama 2 7B, which is comparable to Gemma 2B (Table 6 of [the Gemma paper](https://arxiv.org/pdf/2403.08295) gives scores of 46.9 vs. 45.0 averaged over 18 benchmarks), so **we believe it is not important to our gains over Spirit LM either.**
>
> Though we did not prioritize this ablation (as LM initialization is “free” and benefits were shown by TWIST and Spirit LM), it is an interesting question and new for speech SSMs. **We have trained a SpeechSSM 4m 2B with random initialization.**
>
> [Here is the training plot](https://raw.githubusercontent.com/demo474/ID4403_Demo/refs/heads/master/figures/rebuttal-2b-randominit.png). Despite starting from a higher training loss (expected), it converges to a lower training loss (unexpected)! We suspect that our removal of RoPE may have harmed transfer, as keeping RoPE gives a similar loss curve (see orange). Note **these are audio token losses**; the generated _text_ may be worse. There are also considerations beyond loss; early experiments suggested that RoPE prevented unbounded length generation, a key requirement for our system. **We will validate this ablation and share results during the discussion period if possible.**

---

### Decision · Program_Chairs · 2025-05-01

**Decision:**

Accept (oral)

**Comment:**

**Summary**

The paper designed a speech language model (spoken language model) based on the existing techniques but placed them in a right combination and first time demonstrating the spoken language model which is able to generate consistent long form speech outperforming all prior works. The key components authors placed together are 1) introducing and using Space State Models for the first time for the spoken language models 2) switching back to no positional embedding to have better long form extrapolation 3) initialization from the pretrained LLM 4) semantic tokens to model language and further reuse of acoustic tokens generation via SoundStorm 5) windowed version of tokenization to reduce boundary effects in the long form generation. Authors reuse the prior metrics and benchmarks to compare with prior works but also introduce the long-form variant of the LibriSpeech (and release it) as well as adopt/propose the reference-based semantic metrics, side-by-side LLM-as-judge, and time-stratified evaluation specifically to evaluate the long-form speech.

**Strengths and weaknesses**

All reviewers are supportive of the paper, pointing out the contributions of designing long-form generative speech language model, new benchmark and evaluation protocol, comprehensive evaluation and reasonable approach and evaluation. Some concerns were raised about details of human evaluation, scaling properties, novelty (as SSM is not novel), absence of ablations on the hyperparameters. During rebuttal authors strengthen the paper providing more insights on these all aspects showcasing scalability to the 9B model, extended human evaluation and some insights on the ablations. In the end all reviewers were satisfied with the authors extra details, ablations and clarifications which was reflected in keeping scores on acceptance or increasing the final score for the paper.

**Recommendation**

Given well written paper, strong contributions and comprehensive study, rebuttal from the authors with clarifications and extensive additional details on the results, supportive position from all reviewers, and also importance of the results showing for first time that the spoken language model can be trained for long form speech generation and giving more or less simple recipe to do so, **I strongly recommend acceptance of the paper.**